# Metabolome and Transcriptome Association Analysis Reveals Dynamic Regulation of Purine Metabolism and Flavonoid Synthesis in Transdifferentiation during Somatic Embryogenesis in Cotton

**DOI:** 10.3390/ijms20092070

**Published:** 2019-04-26

**Authors:** Huihui Guo, Haixia Guo, Li Zhang, Zhengmin Tang, Xiaoman Yu, Jianfei Wu, Fanchang Zeng

**Affiliations:** 1State Key Laboratory of Crop Biology, College of Horticulture Science and Engineering, Shandong Agricultural University, Tai’an 271018, China; hhguo@sdau.edu.cn; 2State Key Laboratory of Crop Biology, College of Agronomy, Shandong Agricultural University, Tai’an 271018, China; diya_haixiaguo@163.com (H.G.); 15610418001@163.com (L.Z.); 17861500710@163.com (Z.T.); 15890636826@163.com (X.Y.); jfwu@sdau.edu.cn (J.W.)

**Keywords:** cotton, somatic embryogenesis, transdifferentiation, widely targeted metabolomics, purine metabolism, flavonoid biosynthesis, molecular and biochemical basis, transcript-metabolite network

## Abstract

Plant regeneration via somatic embryogenesis (SE) is a key step during genetic engineering. In the current study, integrated widely targeted metabolomics and RNA sequencing were performed to investigate the dynamic metabolic and transcriptional profiling of cotton SE. Our data revealed that a total of 581 metabolites were present in nonembryogenic staged calli (NEC), primary embryogenic calli (PEC), and initiation staged globular embryos (GE). Of the differentially accumulated metabolites (DAMs), nucleotides, and lipids were specifically accumulated during embryogenic differentiation, whereas flavones and hydroxycinnamoyl derivatives were accumulated during somatic embryo development. Additionally, metabolites related to purine metabolism were significantly enriched in PEC vs. NEC, whereas in GE vs. PEC, DAMs were remarkably associated with flavonoid biosynthesis. An association analysis of the metabolome and transcriptome data indicated that purine metabolism and flavonoid biosynthesis were co-mapped based on the Kyoto encyclopedia of genes and genomes (KEGG) database. Moreover, purine metabolism-related genes associated with signal recognition, transcription, stress, and lipid binding were significantly upregulated. Moreover, several classic somatic embryogenesis (SE) genes were highly correlated with their corresponding metabolites that were involved in purine metabolism and flavonoid biosynthesis. The current study identified a series of potential metabolites and corresponding genes responsible for SE transdifferentiation, which provides a valuable foundation for a deeper understanding of the regulatory mechanisms underlying cell totipotency at the molecular and biochemical levels.

## 1. Introduction

Somatic embryogenesis (SE) plays a crucial role in the genetic transformation of plants and in their in vitro rapid propagation. SE refers to the process of transformation from the somatic to the embryogenic state and is a unique phenomenon similar to what occurs during the development of zygotic embryos. The SE process can be artificially controlled in vitro conditions. It is also a classic example of cell totipotency. The underlying mechanisms of SE are significant for revealing important scientific theoretical problems that involve cell development, differentiation, and morphogenesis [1,2,3,4]. 

Nic-Can et al. [5] reported that SE is a complex process of molecular regulation in which somatic cells acquire totipotency to transform into embryonic cells. How does a somatic cell become a whole plant? This question has been reported to be one of the 25 questions most important within the scientific community [6]. However, the regulatory networks that are involved in the transition from the nonembryogenic staged callus (NEC) to the somatic embryo during SE remain poorly understood [7]. In addition to environmental factors, external stimuli and hormones, many specifically expressed genes also participate in the SE process and play a decisive role [8,9,10,11]. Therefore, it is particularly important to identify and isolate key genes that regulate SE. Candidate genes that may regulate SE were identified using bioinformatics tools [12]. The results showed that *auxin response factor* (*ARF*) [11,13,14], *leafy cotyledon* (*LEC*) [10,15], *Wuschel* (*WUS*) [16,17,18], *somatic embryogenesis receptor kinase* (*SERK*) [8], *shoot meristemless* (*STM*), and *baby boom* (*BBM*) [10,19,20] were involved in SE. In addition, the salicylic acid (SA) and jasmonic acid (JA) signaling pathways were also predicted to regulate SE [12,21,22,23,24,25,26,27]. 

During the process of cotton SE, Hu et al. [28] found that interference of the high-mobility group box 3 (*GhHmgb3*) gene enhanced the proliferation and differentiation of embryogenic callus. Poon et al. [29] found that embryogenic callus could secrete a specific AGP protein (GhPLA1), which was extracted and added to the medium to enhance SE. Min et al. [7] showed that the process of SE was negatively regulated by a casein kinase gene (*GhCKI*) via a complex regulatory network. Several differentially expressed small RNAs and their target genes were identified via small RNA sequencing and degradation sequencing, which indicated that the SE process in cotton was regulated by complex gene expression networks [30]. Genes related to stress, such as *SERK1*, *abscisic aldehyde synthesis enzyme 2* (*ABA2*), *abscisic acid insensitive 3* (*ABI3*), *jasmonate ZIM-domain 1* (*JAZ1*), *late embriogenesis abundant protein 1* (*LEA1*), and transcription factors (NACs, WRKYs, MYBs, ERFs, Zinc finger family proteins) were also involved in callus induction [31,32,33,34,35,36,37]. Transcriptome sequencing was carried out during callus dedifferentiation and redifferentiation. Auxin and stress response molecules were found to be involved in SE in cotton [38]. Xu et al. [39] also found that plant growth regulators played an important regulatory role in SE.

In recent years, metabolomics, which involves the collection of all low molecular weight metabolites that are present in a certain organism, cell, or tissue during a specific physiological period, has attracted a great deal of attention. It mainly involves the examination of endogenous small molecules with molecular weights less than 1000 Da. Plants can produce 200,000 to 1 million kinds of metabolites, which can be divided into primary metabolites and secondary metabolites [40].

Metabolites are the ultimate result of gene transcription and protein expression in organisms that are under the influence of internal and external factors. They form a bridge between genes and phenotypes and can directly reflect the physiological phenomena within plants. At the same time, metabolites can regulate gene transcription and protein expression. As the final products of cellular regulatory processes, metabolites represent the building blocks of macromolecules and are also an essential component of cellular energy pathways [41]. Metabolomic technologies enable the examination and identification of endogenous biochemical reaction products and thereby reveal information about the metabolic pathways and processes occurring within a living cell [42]. 

Metabolomics data can provide a wealth of information about the biochemical status of tissues, and the interpretation of such data offers an effective approach that can be used for the functional characterization of genes [43,44,45]. Transcriptomic and proteomic studies are only able to predict changes at the gene expression and protein levels, respectively, while metabolomics studies investigate the changes in functioning exhibited by these genes or proteins [46]. Compared with the genome, transcriptome, or proteome, the metabolome more accurately reflects the phenotype of the organism, and the minor changes in the genome and proteome can be reflected and amplified by the metabolome [47]. The applicability of metabolomics to the SE process has been demonstrated in many plant genera [48,49,50,51,52].

The development of ‘omics’ technology has allowed for comprehensive analysis of the SE process at the transcript, protein, and metabolite levels [50,52]. The combination of modern omics technologies such as transcriptomics and metabolomics provides a great opportunity to acquire a deeper understanding of the mechanisms of cell totipotency at the molecular and biochemical levels [52]. To date, there have been few studies that have used transcriptomics and metabolomics techniques to identify potential key factors involved in the SE process [48,49,51,52]. Additionally, the relationship between callus tissues and the media used to culture them is not well understood at the biochemical level in terms of nutrient uptake by callus from the medium or release of chemicals into the medium [52]. Integrated metabolomic and transcriptomic network analyses can elucidate the functioning of a series of secondary metabolites along with changes in content, as well as the corresponding differentially expressed genes, which can broaden the global view of SE regulation [52]. 

In this study, we comparatively investigated the dynamic metabolomic and transcriptomic profiles of two transdifferentiation processes, embryogenic differentiation and somatic embryo development, during the SE process in cotton. By interactively comparing metabolomic and transcriptomic data, we were able to identify potential metabolites and the corresponding differentially expressed genes at the molecular and biochemical levels, as well as draw a comprehensive picture of the events underpinning SE development to provide a valuable foundation for uncovering the regulatory mechanisms underlying plant cell totipotency.

## 2. Results

### 2.1. Transdifferentiation Staged Cultures Derived from Cotton Somatic Embryogenesis

We established an efficient cotton SE system and obtained cultures from different developmental stages, including nonembryogenic staged calli (NEC), primary embryogenic calli (PEC), and initiation staged embryos with globular-like enriched (GE) based on our previous approach as published recently [53] (Figure 1). These representative staged samples would be highly enriched and collected to establish the metabolic and transcriptional profiles.

### 2.2. UPLC-MS/MS-Based Quantitative Metabolomic Analysis and Overall Metabolite Identification

Ultra-performance liquid chromatography (UPLC) and tandem mass spectrometry (MS/MS) were conducted to assess the dynamic metabolite changes in NEC, PEC, and GE in cotton. During the process of instrumental analysis, one quality control (QC) sample was injected after every ten samples to monitor the repeatability of the analysis process. The repeatability of metabolite extraction and detection can be judged using overlapping analysis of the total ion current (TIC) in the different QC samples (Figure 2). The results showed the overlap in the TIC curves during metabolite detection. The retention times and peak intensities were consistent, which indicated that the signal was stable when an identical sample was detected at a different time. The instrumental stability provided an important guarantee of the repeatability and reliability of our metabonomic data. 

The determination of Pearson’s correlation coefficient reflected the repeatability among the intragroup samples (Figure 3a). Principal component analysis (PCA) was used to determine the rate of contribution of the first two primary components (67.49%). The three period materials were obviously separated, and each formed a cluster (Figure 3b). These results suggested that there was sufficient reproducibility of the materials, which made them suitable for the following qualitative and quantitative analyses.

After quality validation, a total of 581 metabolites with known structures were identified in NEC, PEC and GE, each of which was analyzed using three biological replicates (Appendix A). Of the 581 metabolites, amino acids (15%), flavones (15%), organic acids (12%), lipids (11%), and nucleotides (10%) accounted for a large proportion (Figure 3c). Detailed information about the identified metabolites, including the compounds, classes, molecular weights (Da), ionization models, Kyoto Encyclopedia of Genes and Genomes (KEGG) pathways, and quantities for each of the three periods is shown in Appendix A.

### 2.3. Identification of Differentially Accumulated Metabolites (DAMs)

Differentially accumulated metabolites (DAMs) were defined as those exhibiting a fold change ≥2 or a fold change ≤0.5 and a variable importance in project (VIP) ≥1 between PEC versus NEC, GE versus PEC, or GE versus NEC (*p* < 0.05). In total, 156, 139, and 159 DAMs were identified, respectively (Table 1 and Appendix A). For PEC vs. NEC, 124 of 156 (79.5%) were upregulated and 32 of 156 (20.5%) were downregulated. Of the 139 DAMs identified between GE and PEC, 86 (61.9%) and 53 (38.1%) metabolites were upregulated and downregulated, respectively. Of the 159 metabolites differentially accumulated in GE compared to NEC, 128 (80.5%) and 31 (19.5%) metabolites were upregulated and downregulated, respectively. Volcano plots were generated to represent the significant differences between PEC vs. NEC, GE vs. PEC, and GE vs NEC (Figure 4a–c). A hierarchical cluster analysis was also performed to assess the DAM accumulation patterns (Figure 4d–f).

Of these DAMs, amino acids (17%), organic acids (13%), flavones (12%), nucleotides (12%), and hydroxycinnamoyl derivatives (6%) accounted for a large proportion in PEC compared to NEC (Figure 4g). Flavones (23%), amino acids (12%), hydroxycinnamoyl derivatives (9%), organic acids (9%), and other compounds (9%) were shown to be differentially accumulated in GE and PEC (Figure 4h). Of the differentially accumulated metabolites in GE vs. NEC, flavones (16%), organic acids (15%), amino acids (14%), nucleotides (11%), and other metabolites (8%) had the highest representation (Figure 4i). Differential metabolites commonly and specifically accumulated in PEC vs. NEC, GE vs. PEC, and GE vs. NEC were also investigated (Figure 5).

### 2.4. Functional Annotation and KEGG Enrichment Analysis of DAMs

Differentially accumulated metabolites in PEC vs. NEC, GE vs. PEC, and GE vs. NEC were functionally annotated using the Kyoto Encyclopedia of Genes and Genomes (KEGG) database, which was summarized in Appendix A. To further investigate the functioning of SE-related metabolites, we analyzed the differences and dynamic changes of biological processes among the three groups. For PEC vs. NEC (embryogenic differentiation), the KEGG enrichment analysis showed that the terms ‘purine metabolism’, ‘cGMP-PKG signaling pathway’, and ‘olfactory transduction’ were significantly enriched (Figure 6a). However, the terms ‘phenylpropanoid biosynthesis’, ‘flavone and flavonol biosynthesis’, ‘flavonoid biosynthesis’, and ‘biosynthesis of phenylpropanoids’ were enriched in GE vs. PEC (somatic embryo development) (Figure 6b). Meanwhile, for GE vs. NEC, the DAMs were most strongly associated with the terms ‘purine metabolism’, ‘microbial metabolism in diverse environments’, ‘biosynthesis of plant secondary metabolites’, and ‘cGMP-PKG signaling pathway’ (Figure 6c). 

### 2.5. Association Analysis of the Metabolome and Transcriptome Data Sets

To provide an overview of the system-wide changes that occur during SE in cotton, transcriptomic profiling based on RNA-seq was performed. Significant differentially expressed genes (DEGs) were detected among NEC, PEC, and GE (unpublished data). By interactively comparing the metabolomic and transcriptomic data, we were able to identify potential metabolites and the corresponding differentially expressed genes at the molecular and biochemical levels.

#### 2.5.1. KEGG Enrichment Analysis of DAMs and DEGs During Cotton SE

To investigate the relationships between genes and metabolites involved in the two transdifferentiation processes during SE in cotton, the DAMs and DEGs in the two compared groups (PEC vs. NEC and GE vs. PEC) were mapped according to the KEGG database. There were 96 and 55 co-mapped pathways in PEC vs. NEC and GE vs. PEC, respectively (Appendix A).

Interestingly, of these co-mapped pathways, ‘purine metabolism’ (ko00230) was significantly enriched in PEC vs. NEC. Meanwhile, ‘phenylpropanoid biosynthesis’ (ko00940), ‘flavonoid biosynthesis’ (ko00941), and ‘flavone and flavonol biosynthesis’ (ko00944) showed significant accumulation in GE vs. PEC. DAMs involved in purine metabolism during embryogenic differentiation and in flavonoid biosynthesis during somatic embryo development are listed in Table 2 and Table 3, respectively.

#### 2.5.2. Correlation Analysis of DAMs and DEGs during SE in Cotton

Log_2_ conversion data for the differentially accumulated metabolites and differentially expressed genes were selected that had a Pearson’s correlation coefficient (PCC) > 0.8. To obtain a systematic view of the variations in metabolites and their corresponding genes with PCC > 0.8, nine quadrant diagrams were generated for the embryogenic differentiation and somatic embryo development processes (Figure 7). 

The black dotted lines divide each graph into 9 quadrants. Upregulated metabolites and downregulated genes are displayed in quadrant 1, upregulated metabolites and unchanged genes are displayed in quadrant 2, upregulated metabolites and upregulated genes are displayed in quadrant 3, unchanged metabolites and downregulated genes are displayed in quadrant 4, unchanged metabolites and unchanged genes are displayed in quadrant 5, unchanged metabolites and upregulated genes are displayed in quadrant 6, downregulated metabolites and downregulated genes are displayed in quadrant 7, downregulated metabolites and unchanged genes are displayed in quadrant 8, and downregulated metabolites and upregulated genes are displayed in quadrant 9. Moreover, the DAMs and DEGs shown in quadrant 3 and quadrant 7 are positively correlated and have similar consistent patterns, while the DAMs and DEGs shown in quadrant 1 and quadrant 9 are negatively correlated and have opposite patterns. 

Notably, in quadrant 3, several representative factors that are related to receptor-like protein kinases, signal recognition, transcription, stress regulation, lipid binding, hormone responses, and histone modification were significantly activated during embryogenic differentiation process (Table 4).

#### 2.5.3. Transcript-Metabolite Correlation Network Representing DAMs and DEGs during SE in Cotton

To model the synthetic and regulatory characteristics of DAMs and DEGs, subnetworks were constructed to determine transcript-metabolite correlations. Only the correlation pairs with a correlation coefficient > 0.8 were included in the analysis.

In PEC vs. NEC, the differential metabolites involved in purine metabolism are listed in Table 2. Several classic key genes that have been shown to be specifically activated during embryogenic differentiation were included in the correlation test, including *BBM* [10,19,20], *SERK* [8], *LEC* [10,15], *ARF* [11,13,14], *AGL15* [54], *AGP* [29], *GLP* [55], *AGO1* [56], *LTP*, and *AMY* [57]. The visualized network revealed that a total of 22 nodes were connected by 81 edges. Meanwhile, 53 pairs showed a positive correlation, and 28 pairs were negatively correlated (Figure 8a). Detailed information about the gene-metabolite pairs that are involved in purine metabolism during embryogenic differentiation is listed in Table 5.

The differential metabolites involved in flavonoid biosynthesis in GE vs. PEC are listed in Table 3. Genes that control somatic embryo development, including *wuschel* (*WUS*) [16,17,18], *clavata1* (*CLV1*) [16,17,18], *cup-shaped cotyledon 2* (*CUC2*) [58], *short-root* (*SHR*), and *scarecrow* (*SCW*) [59], were subjected to correlation tests. A transcript-metabolite correlation network was built that consisted of 12 nodes and 13 edges. Four pairs showed a positive correlation, and nine pairs were negatively correlated (Figure 8b). Detailed information about the gene-metabolite pairs involved in flavonoid biosynthesis during somatic embryo development is listed in Table 6.

These results indicated that several classic SE-related genes were highly correlated with their corresponding metabolites involved in purine metabolism and flavonoid biosynthesis, which reconfirmed the specific importance of purine metabolism during embryogenic differentiation, as well as the importance of flavonoid biosynthesis during somatic embryo development. The transcriptome data validated the authenticity and accuracy of the metabolic analysis.

## 3. Discussion

Metabolomics is an emerging omics technology that, like genomics and proteomics, can be used to qualify and quantify all metabolites of small molecular weight within the cells of an organism. Plant metabolomics has been widely applied to the investigation of patterns of metabolite accumulation and their underlying genetic basis via the identification of genes involved in metabolism, which is currently a topic of interest in modern plant biology. As the final products of genome expression, metabolites directly define the biochemical characteristics of a cell or tissue, which allows metabolomic data to be used to explain the biochemical mechanisms underlying SE. Meanwhile, integrated transcriptomic and metabolomic analysis allows for the more precise representation of gene-to-metabolite networks and thus will be an effective method that can be used to decipher the mechanisms involved in SE regulation in cotton. In the current study, we combined transcriptome and metabolome analyses to generate dynamic profiles of two transdifferentiation processes, embryogenic differentiation, and somatic embryo development, during SE in cotton, with the aim to provide a better understanding of the processes of SE transdifferentiation that resulted in cell totipotency at the molecular and biochemical levels.

### 3.1. Accumulated DAMs Specifically Involved in Two SE Transdifferentiation Processes in Cotton

In this study, a hierarchical cluster analysis (HCA) was performed to assess the patterns of accumulation of the metabolites among the different samples. Results showed that PEC and GE can be clustered together while NEC forms a separate cluster, which suggests that there is significant differential accumulation of metabolites between the two transdifferentiation processes, embryogenic differentiation (PEC-NEC) and somatic embryo developmental initiation (GE-PEC).

To investigate the differential metabolites specifically accumulated in (PEC-NEC) vs. (GE-PEC) vs. (GE-NEC), a Venn diagram was generated (Figure 5a; Appendix A). The results showed that 26 metabolites were accumulated in common among (PEC-NEC) vs. (GE-PEC) vs. (GE-NEC). Meanwhile, 45 DAMs were accumulated in both (PEC-NEC) vs. (GE-PEC), 118 DAMs in both (PEC-NEC) vs. (GE-NEC), and 63 DAMs in both (GE-PEC) vs. (GE-NEC). 

We examined the metabolites that were specifically accumulated between the two transdifferentiation processes, embryogenic differentiation (PEC-NEC), and somatic embryo development (GE-PEC). The results showed that there were 111 DAMs that were specifically accumulated in PEC vs. NEC, of which amino acids, organic acids, nucleotides, flavones, and lipids were the most represented (Figure 5b). Meanwhile, 94 DAMs were accumulated specifically in GE vs. PEC, of which flavones hydroxycinnamoyl derivatives, other metabolites, amino acids and organic acids accounted for a large proportion (Figure 5c). Specifically, nucleotides and lipids were more greatly accumulated during embryogenic differentiation, whereas greater amounts of flavones and hydroxycinnamoyl derivatives were accumulated during the somatic embryo development process. These data indicated that nucleotides and lipids may play important and special roles during embryogenic differentiation, whereas flavonoids are more important during the embryo maturation process.

### 3.2. Enrichment of Purine Metabolism in Embryogenic Differentiation

Purine metabolism refers to the metabolic pathways that synthesize and break down the purines that are present in many organisms. In our study, to investigate the functioning of SE-related metabolites, we analyzed the differences and dynamic metabolite changes among the three groups, PEC vs. NEC, GE vs. PEC, and GE vs. NEC. KEGG enrichment analysis showed that the ‘purine metabolism’ pathway was significantly enriched during embryogenic differentiation (Figure 6b). All of the differentially accumulated metabolites related to purine metabolism were upregulated. Meanwhile, enrichment analysis of DAMs and DEGs showed that ‘purine metabolism’ was co-mapped based on results from the KEGG database (Appendix A). 

In our study, several major DEGs of regulatory factors were identified and significantly associated with purine metabolism in embryogenic differentiation, including receptor-like protein kinases, signal recognition, transcription, stress regulation, lipid binding, hormone responses, and histone modification were significantly upregulated during the embryogenic differentiation process (Table 4). 

The role of purine metabolism in the SE process has been demonstrated in many plants. A comparative omic analysis of tree fern *Cyathea delgadii* explants that were undergoing direct SE was performed. The results revealed that the differentially regulated proteins adenine phosphoribosyl transferase 3 and adenosine kinase 2 were assigned to the purine metabolism category and were associated with direct SE in *C. delgadii* [60]. To understand the molecular mechanisms that regulate early SE in *Eleutherococcus senticosus* Maxim, a high-throughput RNA-seq technology was used to investigate its transcriptome. The initiation of SE affected gene expression in many KEGG pathways but predominantly affected expression in metabolic pathways and those related to the biosynthesis of secondary metabolites. Other unigenes were classified into the purine metabolism pathway [61]. Similar results were obtained in cotton (*Gossypium hirsutum* L.). RNA-seq was performed to analyze the genes expressed during SE and their expression dynamics using RNAs isolated from nonembryogenic callus (NEC), embryogenic callus (EC), and somatic embryos (SEs). The differentially expressed genes in NEC, EC, and SEs were identified, annotated, and classified. Partial DEGs were identified that were related to purine metabolism [62]. A possible critical role for purines during embryogenesis in geranium hypocotyl tissues (*Pelargonium* x *hortorum*) has also been reported [63]. The above results revealed the significant role of purine metabolism in EC proliferation. 

### 3.3. Enrichment of Flavonoid Biosynthesis in Somatic Embryo Developmental Initiation

Flavonoids are synthesized by the phenylpropanoid metabolic pathway in which the amino acid phenylalanine is used to produce 4-coumaroyl-CoA. The metabolic pathway continues through a series of enzymatic modifications to yield flavanones, dihydroflavonols, and then anthocyanins. Along this pathway, many products can be formed, including flavonols, flavan-3-ols, proanthocyanidins (tannins) and a host of other various polyphenolics. In the current study, 139 differentially accumulated metabolites were identified in GE vs. PEC, and flavone was largely detected (Figure 4h). 94 DAMs accumulated specifically in somatic embryo development, of which flavone accounted for a large proportion (Figure 5c). KEGG enrichment analysis of DAMs suggested that ‘phenylpropanoid biosynthesis’, ‘flavone and flavonol biosynthesis’, ‘flavonoid biosynthesis’, and ‘biosynthesis of phenylpropanoids’ were enriched in GE vs. PEC (Figure 6b). Meanwhile, in co-mapped pathways between genes and metabolites, ‘phenylpropanoid biosynthesis’ (ko00940), ‘flavonoid biosynthesis’ (ko00941), and ‘flavone and flavonol biosynthesis’ (ko00944) showed significant accumulation in GE vs. PEC (Appendix A). 

The significant role of flavonoid biosynthesis in the somatic embryo development process has been investigated. In citrus cell cultures, there was no detectable accumulation of flavonoid in the undifferentiated calli, but flavonoid accumulated after the morphological changes to embryos. Two chalcone synthase (*CHS*) genes differentially expressed during citrus SE and *CHS* gene may regulate the accumulation of flavonoid [64].

With the goal to better understand SE development and to improve the efficiency of SE conversion in *Theobroma cacao* L., gene expression differences between zygotic and somatic embryos were examined using a whole genome microarray. Expression levels of genes involved in fatty acid metabolism and flavonoid biosynthesis were differentially expressed in the two types of embryos. The relatively higher expression of flavonoid related genes during SE suggested that the developing tissues may be experiencing high levels of stress during SE maturation caused by the in vitro environment [65].

Moreover, Wang et al. reported that overexpression of *GhSPL10*, a target of *GhmiR157a*, activated the flavonoid biosynthesis pathway, and promoted initial cellular dedifferentiation and callus proliferation [66]. The synthesis of flavonoids like anthocyanin in several plant tissues has been associated with increased phenylalanine ammonia lyase (PAL) activity. The increased PAL level was detected in samples collected from different growth phases during SE in *Silybum marianum*. As intermediary products of phenylpropanoid metabolism, flavonoids may stimulate differentiation and create a situation that is more favorable for embryogenesis [67]. These conclusions demonstrated that flavonoids biosynthesis might be frequently associated with somatic embryo development during cotton SE transdifferentiation.

## 4. Materials and Methods

### 4.1. Plant Materials and Culture Conditions

Upland cotton (*Gossypium hirsutum* cv. YZ-1) seeds were sterilized in 0.1% HgCl_2_ (*w*/*v*) for 8 min and were then rinsed 3–4 times with distilled water. The seeds were then germinated in Murashige and Skoog (MS) medium supplemented with 3% (*w*/*v*) sucrose and 0.25% (*w*/*v*) phytagel. Hypocotyl explants (0.5–1.0 cm) from 7-day-old seedlings were cultured in MS plus B_5_ vitamin (MSB) medium containing 0.45 μmol·L^−1^ 2,4-dichlorophenoxyacetic acid (2,4-D) and 0.46 μmol·L^−1^ kinetin (KT). NEC were maintained in MSB medium for 6 weeks at 28 °C in a 16/8 h light/dark photoperiod and were then subcultured in fresh MSB medium without hormones. Following an additional 3–4 weeks of growth, the somatic-to-embryogenic transition progressed to the point where it induced the development of PEC and GE. Based on our previous approach as published recently [53], these representative staged samples of NEC, PEC, and GE could be highly enriched and collected, frozen immediately in liquid nitrogen, and stored at −80 °C for subsequent metabolic and transcriptomic profiling. The sample from each stage was prepared using three biological replicates.

### 4.2. Sample Preparation and Extraction for Widely Targeted Metabolic Profiling

Chemical extraction was carried out on nine samples (three biological replicates for each of three developmental stages) for the purposes of metabolic analyses. Each freeze-dried sample was crushed using a mixer mill (MM 400, Retsch, Haan, Germany) with a zirconia bead for 1.5 min at 30 Hz. One hundred milligrams of powder was weighed and extracted overnight at 4 °C in 1.0 mL 70% aqueous methanol. Following centrifugation at 10000 g for 10 min, the extracts were absorbed using a CNWBOND Carbon-GCB SPE Cartridge (250 mg, 3 mL; ANPEL, Shanghai, China, www.anpel.com.cn/cnw) and filtered with a SCAA-104 filter (0.22 μm pore size; ANPEL, Shanghai, China, http://www.anpel.com.cn/) prior to LC-MS analysis.

### 4.3. HPLC Conditions

The sample extracts were analyzed using an LC-ESI-MS/MS system (HPLC: Shim-pack UFLC SHIMADZU CBM30A system, www.shimadzu.com.cn/; MS: Applied Biosystems 6500 Q TRAP, www.appliedbiosystems.com.cn/). The analytical conditions were as follows: HPLC column, Waters ACQUITY UPLC HSS T3 C18 (1.8 μm, 2.1 × 100 mm); solvent system, water (0.04% acetic acid) and acetonitrile (0.04% acetic acid); gradient program, 95:5 *v*/*v* at 0 min, 5:95 *v*/*v* at 11.0 min, 5:95 *v*/*v* at 12.0 min, 95:5 *v*/*v* at 12.1 min, 95:5 *v*/*v* at 15.0 min; flow rate, 0.40 mL/min; temperature, 40 °C; injection volume, 2 μL. The effluent was alternatively connected to an ESI-triple quadrupole-linear ion trap (Q TRAP)-MS. 

### 4.4. ESI-Q TRAP-MS/MS

LIT and triple quadrupole (QQQ) scans were acquired using a triple quadrupole-linear ion trap mass spectrometer (Q TRAP) API 6500 Q TRAP LC/MS/MS System equipped with an ESI Turbo ion-spray interface, operated in positive ion mode and controlled using Analyst 1.6 software (AB Sciex). The ESI source operation parameters were as follows: ion source, turbo spray; source temperature, 500 °C; ion spray voltage (IS), 5500 V; ion source gas I (GSI), gas II (GSII), and curtain gas (CUR), 55, 60, and 25.0 psi, respectively; the collision gas (CAD) was set to high. The instrument tuning and mass calibration were performed using 10 and 100 μmol/L polypropylene glycol solutions in QQQ and LIT modes, respectively. QQQ scans were acquired during MRM experiments in which the collision gas (nitrogen) was set to 5 psi. DP and CE for individual MRM transitions were conducted after further DP and CE optimization. A specific set of MRM transitions were monitored during each period based on the metabolites that eluted during this period [68,69].

### 4.5. Widely Targeted Metabolic Profiling

The samples were analyzed using a metabolomic platform that combined ultra-performance liquid chromatography (UPLC) and tandem mass spectrometry (MS/MS). Metabolite identification was performed using the MWDB metware database (Metware Biotechnology Co., Ltd. Wuhan, China) and other public databases according to standard metabolic operating procedures. 

### 4.6. Statistical Analysis

The metabolite abundances were quantified using the peak areas. The data obtaining from the metabolite profiling were normalized for the principal component analysis (PCA) and partial least squares-discriminant analysis (PLS-DA) [70,71]. Metabolites with significant differences in content were defined as having a variable importance in project (VIP) ≥ 1 and a fold change ≥ 2 or ≤ 0.5. Fisher’s exact test was applied to identify the significant KEGG pathways with a false discovery rate (FDR) < 0.05 [72]. Gene-metabolite pairs with a Pearson’s correlation coefficient > 0.8 were used to construct the transcript-metabolite network.

## 5. Conclusions

Somatic embryogenesis is the developmental reprogramming of somatic cells toward the embryogenesis pathway. In this study, the dynamic metabolomic and transcriptomic profiling of cotton SE transdifferentiation processes, embryogenic differentiation, and somatic embryo development, were comparatively investigated. 

During embryogenic differentiation (PEC vs. NEC), nucleotides and lipids were specifically accumulated. And the metabolome wide DAMs significantly enriched in purine metabolism (Table 2). In addition, purine metabolism-related genes associated with signal recognition, transcription, stress, and lipid binding were remarkably activated (Table 4). Moreover, several classic SE genes that specifically activated during embryogenic differentiation, including *BBM*, *SERK1*, *LEC1*, *ARF2*, *AGL15*, *AGP1*, *GLP2*, *AGO1*, *LTP2*, and *AMY1*, were highly correlated with the corresponding metabolites that were involved in purine metabolism (Table 5).

During somatic embryo development (GE vs. PEC), flavones and hydroxycinnamoyl derivatives were largely accumulated. DAMs were most significantly associated with flavonoid biosynthesis (Table 3). Classic SE genes that control somatic embryo development, including *WUS*, *CLV1*, *CUC2*, *SHR*, and *SCW*, were highly correlated with the corresponding metabolites that were involved in flavonoid biosynthesis (Table 6).

By interactively comparing metabolomic and transcriptomic data, we identified a series of potential metabolites and the corresponding differentially expressed genes candidate for a relevant role in SE transdifferentiation. The findings in our work provide new insights into the underlying molecular and biochemical basis associated with embryogenic competence acquisition underpinning cotton SE development.

## Figures and Tables

**Figure 1 ijms-20-02070-f001:**
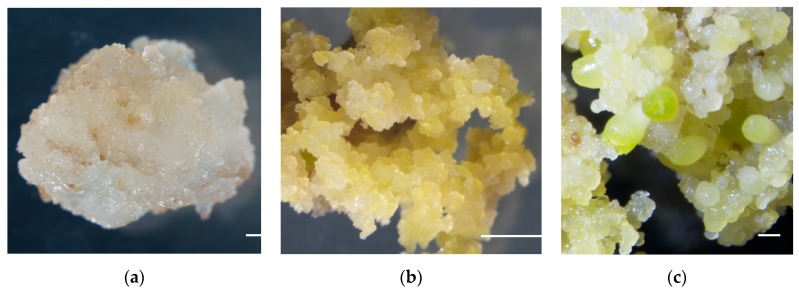
Cultures derived from different developmental stages of cotton somatic embryogenesis (SE) for the purposes of the metabolome and transcriptome assays. (**a**) Nonembryogenic staged calli (NEC); (**b**) primary embryogenic calli (PEC); (**c**) initiation staged embryos with globular-like enriched (GE). Bars = 1 mm.

**Figure 2 ijms-20-02070-f002:**
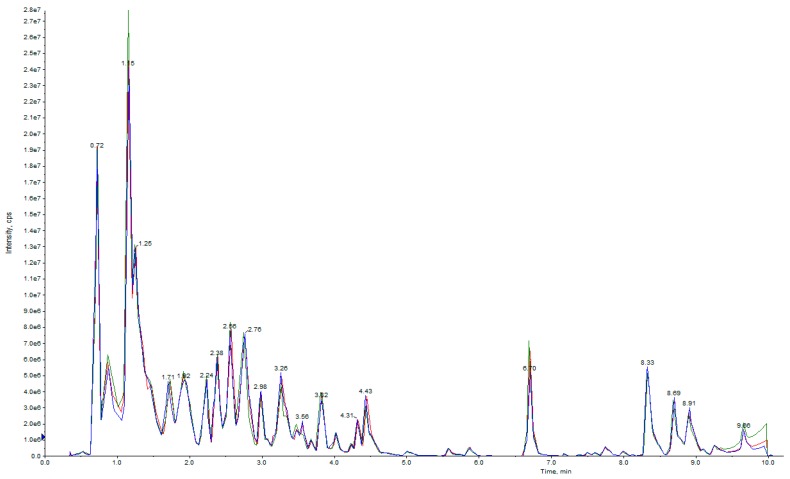
Overlapping analysis of the total ion current (TIC) in different quality control (QC) samples. The abscissa represents the retention time (min) of metabolite detection, and the ordinate represents the intensity of the ion current (cps: count per second).

**Figure 3 ijms-20-02070-f003:**
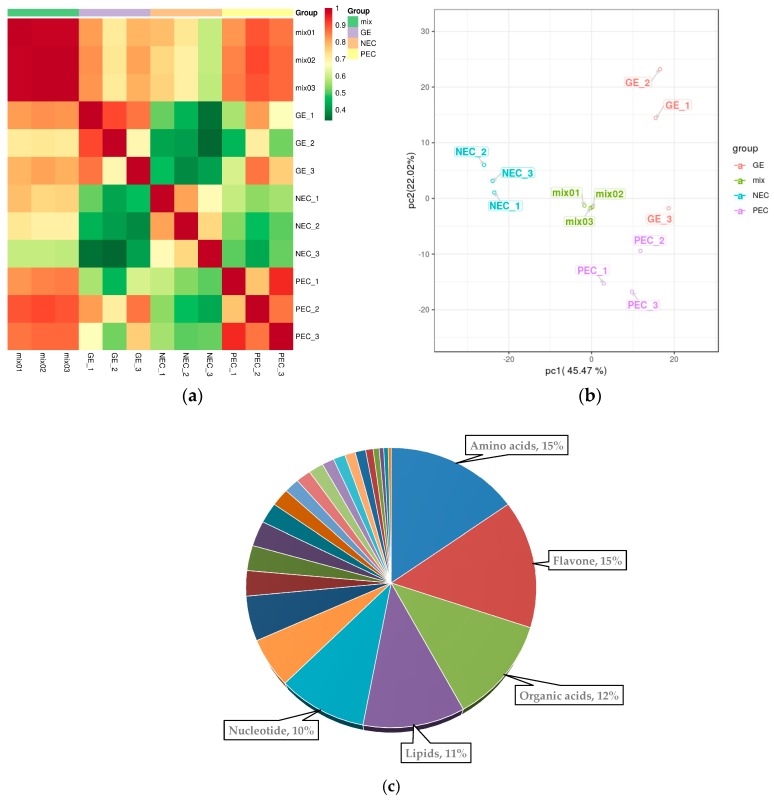
Overall qualitative and quantitative analysis of the metabolomics data. (**a**) Pearson’s correlation coefficients among the three samples (NEC, PEC, and GE) and quality control samples (mix); (**b**) PCA analysis of the three samples (NEC, blue; PEC, purple; GE, red) and quality control samples (mix, green); the x-axis represents the first principal component and the y-axis represents the second principal component. (**c**) Component analysis of the identified metabolites. The top five metabolites are shown beside the graph. NEC, nonembryogenic staged calli; PEC, primary embryogenic calli; GE, initiation staged embryos with globular-like enriched.

**Figure 4 ijms-20-02070-f004:**
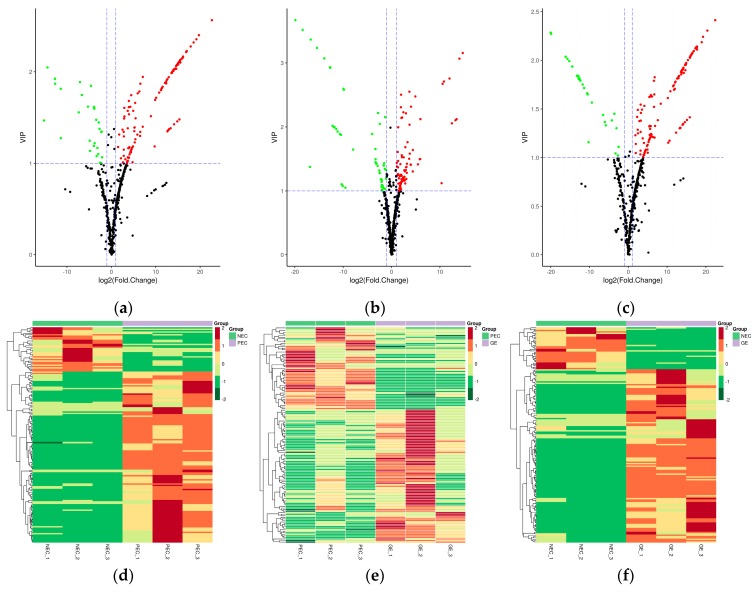
Differentially accumulated metabolites (DAMs) among NEC, PEC, and GE. (**a**) Volcano plot showing the differential metabolites in PEC vs. NEC; (**b**) volcano plot showing the differential metabolites in GE vs. PEC; (**c**) volcano plot showing the differential metabolites in GE vs. NEC. The red spots represent upregulated DAMs, the green dots represent downregulated DAMs, and the black dots represent nondifferentially accumulated metabolites; (**d**) Heat map representing the hierarchical cluster analysis in PEC vs. NEC; (**e**) heat map representing the hierarchical cluster analysis in GE vs. PEC; (**f**) heat map representing the hierarchical cluster analysis in GE vs. NEC. Red indicates high abundance, whereas the relatively low-abundance metabolites are shown in green; (**g**) component analysis of DAMs in PEC vs. NEC; (**h**) component analysis of DAMs in GE vs. PEC; (**i**) component analysis of DAMs in GE vs. NEC. The top five DAMs are noted beside each graph. NEC, nonembryogenic staged calli; PEC, primary embryogenic calli; GE, initiation staged embryos with globular-like enriched.

**Figure 5 ijms-20-02070-f005:**
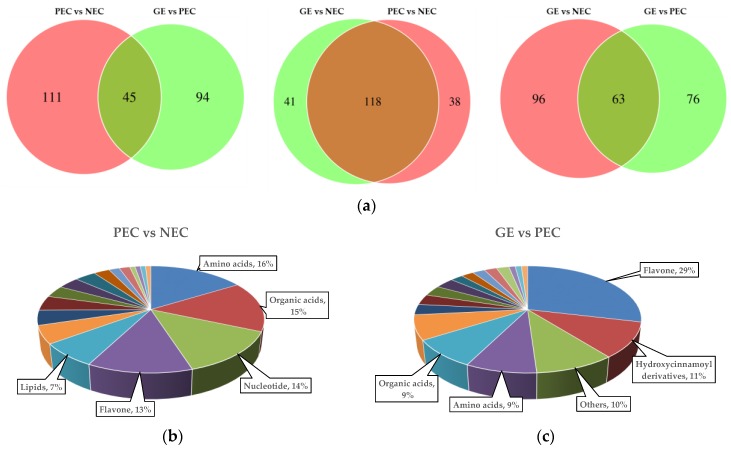
Differential metabolites commonly and specifically accumulated in PEC vs. NEC, GE vs. PEC, and GE vs. NEC. (**a**) Venn diagrams showing the distribution of DAMs in PEC vs. NEC, GE vs. PEC, and GE vs. NEC; (**b**) component analysis of DAMs accumulated specifically in PEC vs. NEC; (**c**) component analysis of DAMs accumulated specifically in GE vs. PEC. The top five specific DAMs are noted beside each graph. DAMs, differentially accumulated metabolites; NEC, nonembryogenic staged calli; PEC, primary embryogenic calli; GE, initiation staged embryos with globular-like enriched.

**Figure 6 ijms-20-02070-f006:**
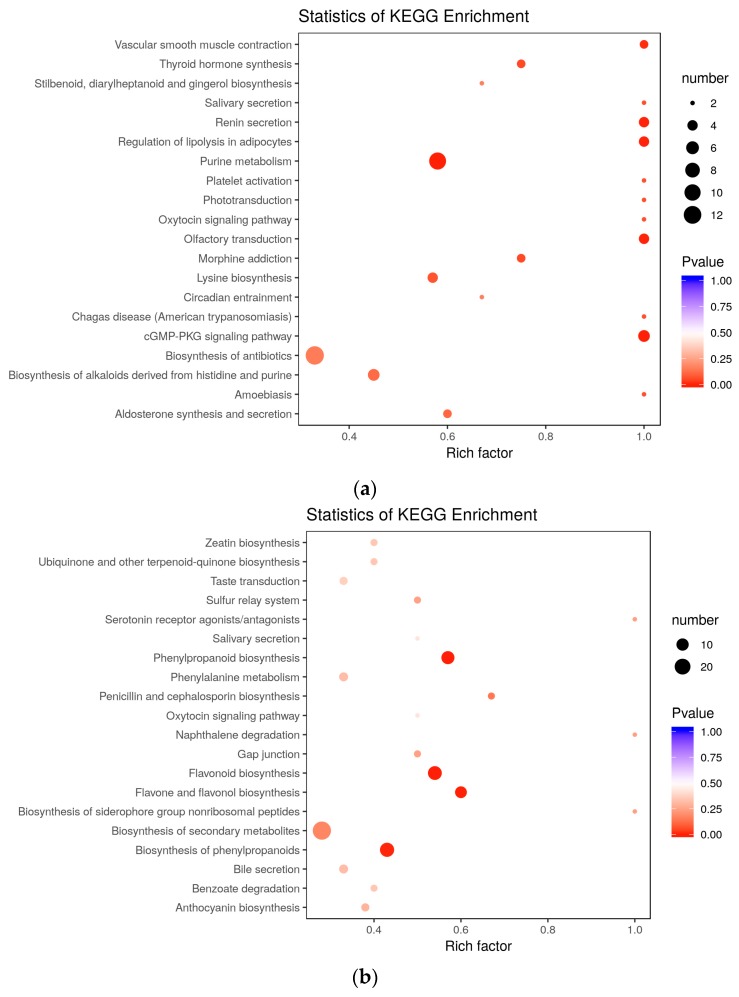
KEGG pathway enrichment analysis of DAMs. (**a**) Pathway enrichment in PEC vs. NEC; (**b**) pathway enrichment in GE vs. PEC; (**c**) pathway enrichment in GE vs. NEC. The x-axis represents the enrichment factor, while the y-axis represents the enrichment pathway. The dot sizes represent the number of differentially enriched metabolites. The statistical analysis of the pathway enrichment was performed using Fisher’s exact test. DAMs, differentially accumulated metabolites; NEC, nonembryogenic staged calli; PEC, primary embryogenic calli; GE, initiation staged embryos with globular-like enriched.

**Figure 7 ijms-20-02070-f007:**
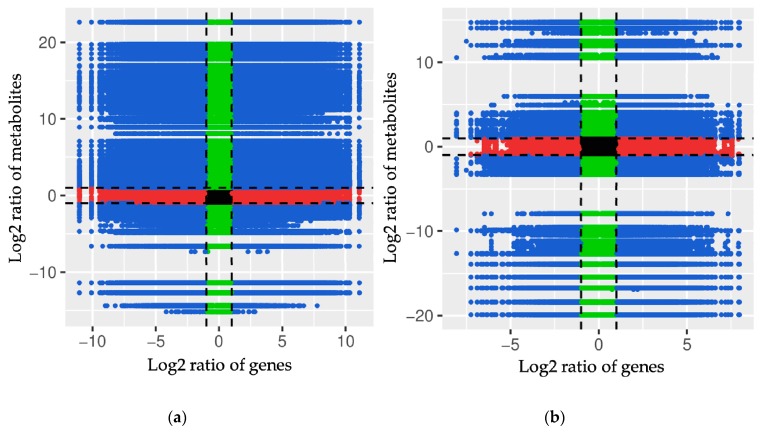
Quadrant diagrams representing association of metabolomic and transcriptomic variation during SE in cotton. (**a**) Overview of metabolomic and transcriptomic variation in PEC vs. NEC; (**b**) overview of metabolomic and transcriptomic variation in GE vs. PEC. The x-axis represents the fold changes in the transcriptome data; the y-axis represents the fold changes in the metabolome data. The black dotted lines represent the differential thresholds. Outside of the threshold lines, there were significant differences in the genes/metabolites, and within the threshold lines are shown the unchanged genes/metabolites. Each point represents a gene/metabolite. Black dots represent the unchanged genes/metabolites; green dots represent differentially accumulated metabolites with unchanged genes; red dots represent differentially expressed genes with unchanged metabolites; blue dots represent both differentially expressed genes and differentially accumulated metabolites. NEC, nonembryogenic staged calli; PEC, primary embryogenic calli; GE, initiation staged embryos with globular-like enriched.

**Figure 8 ijms-20-02070-f008:**
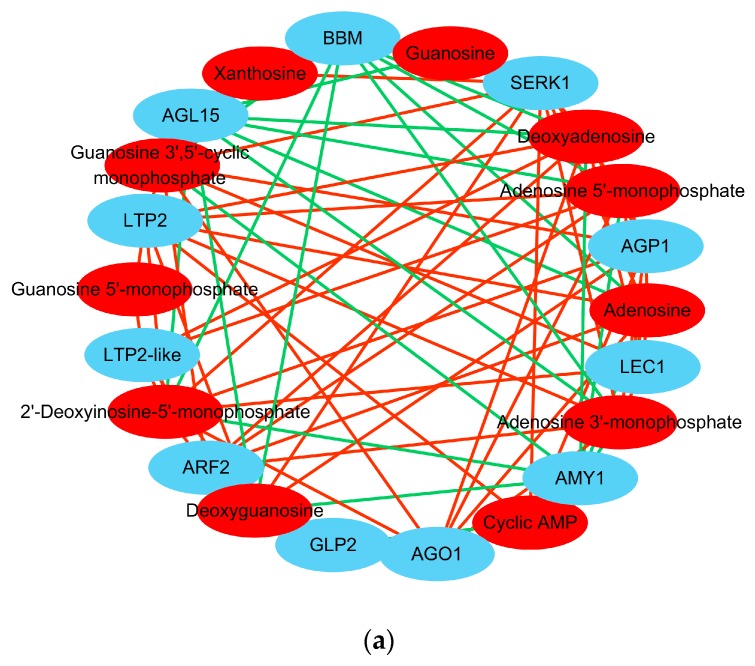
Transcript-metabolite correlation network representing DAMs and DEGs involved in PEC vs. NEC and GE vs. PEC. (**a**) Pearson correlation network representing factors associated with purine metabolism in PEC vs. NEC; (**b**) Pearson correlation network representing factors associated with flavonoid biosynthesis in GE vs. PEC. The gene-metabolite pairs were connected within the network by edges. Blue nodes represent genes and red nodes represent metabolites. Red edges represent positive correlations, and green edges represent negative correlations. DAMs, differentially accumulated metabolites; DEGs, differentially expressed genes; NEC, nonembryogenic staged calli; PEC, primary embryogenic calli; GE, initiation staged embryos with globular-like enriched.

**Table 1 ijms-20-02070-t001:** Summary of differentially accumulated metabolites (DAMs) among NEC, PEC, and GE.

Group Name	Number of Differential Metabolites	Number Upregulated	Number Downregulated
PEC vs. NEC	156	124	32
GE vs. PEC	139	86	53
GE vs. NEC	159	128	31

NEC, nonembryogenic staged calli; PEC, primary embryogenic calli; GE, initiation staged embryos with globular-like enriched.

**Table 2 ijms-20-02070-t002:** DAMs involved in purine metabolism during embryogenic differentiation (PEC vs. NEC).

Index	Compound	Class	Fold Change	Log_2_ (FC)	Type
pmb0981	Adenosine 5′-monophosphate	Nucleotide and its derivates	3.92 × 10^4^	15.3	up
pmb2684	Cyclic AMP	Nucleotide and its derivates	4.79 × 10^4^	15.5	up
pmb2948	Adenosine 3′-monophosphate	Nucleotide and its derivates	2.54 × 10^3^	11.3	up
pmb4344	Guanosine 5′-monophosphate	Nucleotide and its derivates	2.19 × 10^4^	14.4	up
pmc0066	2′-Deoxyinosine-5′-monophosphate	Nucleotide and its derivates	1.29 × 10^4^	13.7	up
pmd0023	Adenosine	Nucleotide and its derivates	1.36 × 10^2^	7.08	up
pme1175	Guanosine	Nucleotide and its derivates	93.1	6.54	up
pme1181	Deoxyguanosine	Nucleotide and its derivates	17.1	4.10	up
pme1296	Xanthosine	Nucleotide and its derivates	1.37 × 10^2^	7.10	up
pme3835	Guanosine 3′,5′-cyclic monophosphate	Nucleotide and its derivates	1.64 × 10^4^	14.0	up
pme3960	Deoxyadenosine	Nucleotide and its derivates	7.32 × 10^4^	16.2	up

DAMs, differentially accumulated metabolites; NEC, nonembryogenic staged calli; PEC, primary embryogenic calli.

**Table 3 ijms-20-02070-t003:** DAMs involved in flavonoid biosynthesis during somatic embryo development (GE vs. PEC).

Index	Compound	Class	Fold Change	Log_2_ (FC)	Type
pme1580	Eriodictyol	Flavanone	4.17	2.06	up
pme2319	Hesperetin	Flavanone	2.43	1.28	up
pmb0605	Apigenin 7-O-glucoside (Cosmosiin)	Flavone	0.145	−2.79	down
pme0088	Luteolin	Flavone	3.56	1.83	up
pme2459	Luteolin 7-O-glucoside (Cynaroside)	Flavone	3.30	1.72	up
pme3300	Tricetin	Flavone	2.60	1.38	up
pme0199	Quercetin	Flavonol	56.9	5.83	up
pme0200	Kaempferol	Flavonol	4.09 × 10^3^	12.0	up
pme0202	Quercetin 3-O-rutinoside (Rutin)	Flavonol	15.9	3.99	up
pme1478	Myricetin	Flavonol	44.2	5.46	up
pme1521	Dihydroquercetin (Taxifolin)	Flavonol	4.60	2.20	up
pme2898	Dihydromyricetin	Flavonol	2.92 × 10^−6^	−18.4	down
pme3212	Quercetin 3-O-glucoside (Isotrifoliin)	Flavonol	6.63	2.73	up
pme3268	Kaempferol 3-O-galactoside (Trifolin)	Flavonol	6.26	2.65	up

DAMs, differentially accumulated metabolites; PEC, primary embryogenic calli; GE, initiation staged embryos with globular-like enriched.

**Table 4 ijms-20-02070-t004:** Significant representative differentially expressed genes (DEGs) involved in purine metabolism during embryogenic differentiation (PEC vs. NEC).

Gene ID	Gene Name	Description	Log_2_ (FC)	Meta ID	Log_2_ (FC)	PCC
Gh_D05G1280	*PERK13*	Proline-rich receptor-like protein kinase	3.61	pme1175	6.54	1
				pme1296	7.10	1
Gh_D10G1867	*IRK*	LRR receptor-like protein kinase	3.52	pmb0981	15.26	1
				pmb2948	11.31	1
				pmc0066	13.65	1
Gh_A03G1831	*CAO*	Signal recognition protein	1.99	pme1175	6.54	1
				pme1296	7.10	1
Gh_A07G2239	*ABI3*	B3 transcription factor	5.70	pmb0981	15.26	1
				pmb2948	11.31	1
				pmc0066	13.65	1
Gh_A08G2488	*WRKY2*	WRKY transcription factor 2	1.77	pmb0981	15.26	1
				pmb2948	11.31	1
				pmc0066	13.65	1
Gh_A11G2618	*NFYC2*	Nuclear transcription factor	3.07	pmb0981	15.26	1
				pmb2948	11.31	1
				pmc0066	13.65	1
Gh_D04G0086	*NFYA3*	Nuclear transcription factor	2.29	pmb0981	15.26	1
				pmb2948	11.31	1
				pmc0066	13.65	1
Gh_A06G1368	*SERP2*	Stress-associated protein	4.36	pme1175	6.54	1
			4.36	pme1296	7.10	1
Gh_D04G1347	*ARR4*	Two-component response regulator	2.55	pme1175	6.54	1
				pme1296	7.10	1
Gh_D12G2180	*AIR1*	Putative lipid-binding protein	3.55	pme1175	6.54	1
			3.55	pme1296	7.10	1
Gh_D12G1336	*HMGB7*	High mobility group B protein	3.75	pmb0981	15.26	1
			3.75	pmb2948	11.31	1
			3.75	pmc0066	13.65	1
Gh_D11G0232	*HMGB13*	High mobility group B protein	5.55	pme1175	6.54	1
			5.55	pme1296	7.10	1
Gh_A06G0239	*IAA9*	Auxin-responsive protein	3.87	pmb2948	11.31	1
Gh_D09G0145	*GASA4*	Gibberellin-regulated protein	6.53	pmb0981	15.26	1
				pmb2948	11.31	1
				pmc0066	13.65	1
Gh_A10G0472	*B34*	Histone H3.2	3.80	pmb0981	15.26	1
			3.80	pmb2948	11.31	1
Gh_A13G1866	*B34*	Histone H3.2	3.46	pmb2948	11.31	1
Gh_D08G1979	*HIS2A*	Histone H2AX	4.15	pmb0981	15.26	1
Gh_D08G0034	*HIS2A*	Histone H2A	3.15	pmb2948	11.31	1

NEC, nonembryogenic staged calli; PEC, primary embryogenic calli.

**Table 5 ijms-20-02070-t005:** Classic gene-metabolite pairs involved in purine metabolism during embryogenic differentiation (PEC vs. NEC).

Gene ID	Gene Name	Description	Log_2_ (FC)	Meta ID	Log_2_ (FC)	PCC
Gh_A08G2008	*BBM*	Baby boom	6.54	pmc0066	13.65	−1
				pmd0023	7.08	−1
				pmb2948	11.31	−1
				pmb0981	15.26	−1
				pme3960	16.20	−1
				pme1181	4.10	−1
				pme1175	6.54	−1
				pme3835	14.00	−1
				pme1296	7.10	−1
Gh_D06G1184	*SERK1*	Somatic embryogenesis receptor kinase	1.25	pmc0066	13.65	1
				pmd0023	7.08	1
				pmb2948	11.31	1
				pmb0981	15.26	1
				pmb2684	15.50	1
				pme3960	16.20	1
				pme1181	4.10	1
				pme1175	6.54	1
				pme3835	14.00	1
				pme1296	7.10	1
Gh_D13G1387	*LEC1*	Leafy cotyledon	6.44	pmc0066	13.65	1
				pmd0023	7.08	1
				pmb2948	11.31	1
				pmb0981	15.26	1
				pme3960	16.20	1
				pme3835	14.00	1
Gh_D07G0476	*ARF2*	Auxin response factor 2	1.89	pmc0066	13.65	1
				pmd0023	7.08	1
				pmb2948	11.31	1
				pmb0981	15.26	1
				pme3960	16.20	1
				pme1181	4.10	1
				pme3835	14.00	1
Gh_A12G0910	*AGL15*	Agamous-like 15	6.47	pmc0066	13.65	−1
				pmd0023	7.08	−1
				pmb2948	11.31	−1
				pmb0981	15.26	−1
				pme3960	16.20	−1
				pme1181	4.10	−1
				pme1175	6.54	−1
				pme3835	14.00	−1
				pme1296	7.10	−1
Gh_D07G0323	*AGP1*	Arabinogalactan protein1	5.93	pmc0066	13.65	1
				pmd0023	7.08	1
				pmb2948	11.31	1
				pmb0981	15.26	1
				pmb2684	15.50	1
				pme3960	16.20	1
				pme1181	4.10	1
				pme3835	14.00	1
Gh_A12G1076	*GLP2*	Germin-like protein 2	−1.13	pmb2684	15.50	−1
				pme1181	4.10	−1
Gh_A09G1557	*AGO1*	Argonaute 1	1.19	pmc0066	13.65	1
				pmd0023	7.08	1
				pmb2948	11.31	1
				pmb0981	15.26	1
				pmb2684	15.50	1
				pme3960	16.20	1
				pme1181	4.10	1
				pme3835	14.00	1
Gh_D05G1937	*AMY1*	Alpha-amylase 1	−4.34	pmc0066	13.65	−1
				pmd0023	7.08	−1
				pmb2948	11.31	−1
				pmb0981	15.26	−1
				pmb2684	15.50	−1
				pme3960	16.20	−1
				pme1181	4.10	−1
				pme3835	14.00	−1
Gh_A12G0504	*LTP2*	Lipid transfer protein 2	3.58	pmc0066	13.65	1
				pmd0023	7.08	1
				pmb2948	11.31	1
				pmb0981	15.26	1
				pmb2684	15.50	1
				pme3960	16.20	1
				pme1181	4.10	1
				pme3835	14.00	1
				pmb4344	14.40	1
Gh_D12G0517	*LTP2-like*	Lipid transfer protein 2-like	3.03	pmc0066	13.65	1
				pmb0981	15.26	1
				pme3960	16.20	1
				pme1181	4.10	1
				pmb4344	14.40	1

NEC, nonembryogenic staged calli; PEC, primary embryogenic calli.

**Table 6 ijms-20-02070-t006:** Classic gene-metabolite pairs involved in flavonoid biosynthesis during somatic embryo development (GE vs. PEC).

Gene ID	Gene Name	Description	Log_2_ (FC)	Meta ID	Log_2_ (FC)	PCC
Gh_A10G0884	*WUS*	Wusche l	2.12	pme3268	2.65	−1
Gh_A02G0853	*CLV1*	Clavata 1	3.00	pme2898	−18.40	1
				pme0200	12.00	−1
				pme3268	2.65	−1
				pme2459	1.72	−1
				pme0202	3.99	−1
Gh_D01G0448	*CUC2*	Cup-shaped cotyledon 2	2.77	pme2459	1.72	−1
				pme3212	2.73	−1
Gh_D02G0017	*SCW*	Scarecrow	2.55	pme2898	−18.40	−1
				pme0200	12.00	1
				pme3268	2.65	1
				pme2459	1.72	1
Gh_A12G0710	*SHR*	Short-root	1.90	pmb0605	−2.79	−1

PEC, primary embryogenic calli; GE, initiation staged embryos with globular-like enriched.

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
