# Peer review of "Metabolome and Transcriptome Association Analysis Reveals Dynamic Regulation of Purine Metabolism and Flavonoid Synthesis in Transdifferentiation during Somatic Embryogenesis in Cotton"

_ijms, 2019, doi:10.3390/ijms20092070_

Round 1
Reviewer 1 Report
The review on the publication under the title: Metabolome and transcriptome association analysis reveals dynamic regulation of purine metabolism and flavonoid synthesis in transdifferentiation during somatic embryogenesis in cotton.
Here my comments:
According to journal requirements the abstract have to 200 words, the authors have much more. Please reduce the length of the abstract demonstrating the main findings in the manuscript.
Line 49 Authors mentioned about genetic transformation of only crops? Is it true only for crops?
Line 52 Authors: The SE process can be artificially controlled in in vitro conditions. More information on this sentence?
Line 60 Authors: in the transition from the nonembryogenic callus to the somatic embryo during SE. Could you please explain me how the NEC became an embryogenic? That’s why this callus called NEC because there is no embryogenic potential and no embryogenic/meristematic cells.
Line 65-66 Authors mentioned about the genes involved in SE processes are these genes the only ones? Any citations? Another thing, please be careful, you used the shortening for Somatic embryogenesis (SE) and on the line 67, you mentioned again the whole name – somatic embryogenesis. Please, check the whole manuscript.
Line 67-67 Any citations about this statement? Examples? I think you could find a lot of them.
Line 79 any examples of the genes involved in the stress?
Line 89 Authors: total between 200,000 and 1 million? Please clarify your statements.
Line 107-117 any citations?
Line 127-130 Authors mentioned about the calli with different morphogenic potential, however I did not understand how the authors separated one callus from another. I do not agree that authors showed as globular embryos. Do you really have only globular embryos, or you have parenchymatous cells as well? Any basic cytology?
Line 138-139, line 252-254 the paragraph show be from more the one sentence. Please correct it.
Line 264-268 These are more conclusions!
Line 360-363 Authors have already mentioned the abbreviations of the genes so there is no point to do that again.
Line 476 Is this Materials and method section?
To sum, the article has the great amount of the results. The discussion is not the great part of it. More information have to be added and properly discussed. Another thing, the basic histological section have to be added to understand the main differences between the analysed calli, as for now, everything is based on the morphological observations. Any conclusions section in the manuscript?
Author Response
Dear Editor and Reviewer:
We are returning our manuscript (ijms-473380) entitled “Metabolome and transcriptome association analysis reveals dynamic regulation of purine metabolism and flavonoid synthesis in transdifferentiation during somatic embryogenesis in cotton” that has been revised based on the reviewer's suggestions. We appreciate the time and expertise of the reviewer and the suggestions helped us to improve our manuscript.
Here are our point by point response to the reviewer's comments:
Reviewer 1
Comments and Suggestions for Authors:
The review on the publication under the title: Metabolome and transcriptome association analysis reveals dynamic regulation of purine metabolism and flavonoid synthesis in transdifferentiation during somatic embryogenesis in cotton.
Here my comments:
1. According to journal requirements the abstract have to 200 words, the authors have much more. Please reduce the length of the abstract demonstrating the main findings in the manuscript.
RESPONSES: We thank and agree with the reviewer’s comments. The abstract has been shortened to make it more concise in the revised manuscript.
2. Line 49 Authors mentioned about genetic transformation of only crops? Is it true only for crops?
RESPONSES: We thank the reviewer for pointing out the confused description here. The term “plants” has been used instead of “crops” in the revised manuscript.
3. Line 52 Authors: The SE process can be artificially controlled in in vitro conditions. More information on this sentence?
RESPONSES: We thank the reviewer for pointing out the mistake. The word “in” has been deleted on this sentence.
4. Line 60 Authors: in the transition from the nonembryogenic callus to the somatic embryo during SE. Could you please explain me how the NEC became an embryogenic? That’s why this callus called NEC because there is no embryogenic potential and no embryogenic/meristematic cells.
RESPONSES: We thank and agree with the reviewer’s comments and suggestions. The term “nonembryogenic callus” is really inappropriate, and has been changed to “nonembryogenic staged callus (NEC)” instead in the entire revised manuscript.
5. Line 65-66 Authors mentioned about the genes involved in SE processes are these genes the only ones? Any citations? Another thing, please be careful, you used the shortening for Somatic embryogenesis (SE) and on the line 67, you mentioned again the whole name-somatic embryogenesis. Please, check the whole manuscript.
RESPONSES: We appreciate the reviewer for the comments and suggestions. These genes are not the only ones involved in SE processes. A series of the relevant references have been cited in the revised manuscript. Other SE related genes, such as BBM, GhHmgb3, GhPLA1 and GhCKI have been referred in the following paragraph. Additionally, we have checked the whole manuscript, and the term “SE” has been used instead of “somatic embryogenesis”.
6. Line 67-67 Any citations about this statement? Examples? I think you could find a lot of them.
RESPONSES: We thank and agree with the reviewer’s comments. A series of the relevant references have been cited in the revised manuscript as the reviewer suggested.
7. Line 79 any examples of the genes involved in the stress?
RESPONSES: Many thanks to the reviewer for the comments and suggestions. The relevant references and examples of genes involved in the stress, such as SERK1, ABA2, ABI3, JAZ1, LEA1 and transcription factors (NACs, WRKYs, MYBs, ERFs, Zinc finger family proteins) have been cited in the revised manuscript.
8. Line 89 Authors: total between 200,000 and 1 million? Please clarify your statements.
RESPONSES: We thank the reviewer for pointing out the confused description here. The sentence has been rewritten to make it clear.
9. Line 107-117 any citations?
RESPONSES: As the reviewer recommend, the relevant references have been cited in the revised manuscript.
10. Line 127-130 Authors mentioned about the calli with different morphogenic potential, however I did not understand how the authors separated one callus from another. I do not agree that authors showed as globular embryos. Do you really have only globular embryos, or you have parenchymatous cells as well? Any basic cytology?
RESPONSES: We thank the reviewer for the valuable comments and suggestions. It is an important point to separate the calli with different morphogenic potential. Actually, we have proposed a feasible way to efficiently distinguish embryogenic and non-embryogenic cells and determine the embryogenic nature of different cotton cell strains. And we established the close associated relationship between morphogenic potential and the corresponding cytological characteristics, as published recently (Guo et al. Identification and characterization of cell cultures with various embryogenic/regenerative potential in cotton based on morphological, cytochemical and cytogenetical assessment [J]. J. Integr. Agr. 2019, 18 (1): 1-8). Based on this approach, the representative staged samples which are feasibly identified by the morphological characteristics, and would be highly enriched and collected in our study. Additionally, we agree with the reviewer’s comments on “globular embryos”, and the term has been changed to “initiation staged embryos with globular-like enriched” in the revised manuscript.
11. Line 138-139, line 252-254 the paragraph show be from more the one sentence. Please correct it.
RESPONSES: We agree with the reviewer’s comments. The two paragraphs have been revised as the reviewer suggested.
12. Line 264-268 These are more conclusions!
RESPONSES: We thank and agree with the reviewer for the comments and suggestions. The sentences have been deleted in the revised manuscript.
13. Line 360-363 Authors have already mentioned the abbreviations of the genes so there is no point to do that again.
RESPONSES: We agree with the reviewer’s comments and suggestions. As the reviewer recommend, the abbreviations of the genes have been used accordingly.
14. Line 476 Is this Materials and method section?
RESPONSES: We thank the reviewer for pointing out the mistake here. “4. Materials and Methods” has been added in the revised manuscript.
15. To sum, the article has the great amount of the results. The discussion is not the great part of it. More information have to be added and properly discussed. Another thing, the basic histological section have to be added to understand the main differences between the analyzed calli, as for now, everything is based on the morphological observations. Any conclusions section in the manuscript?
RESPONSES: We greatly acknowledge and agree with the reviewer’s comments and suggestions. The entire manuscript has been modified as the reviewer recommend. Firstly, we have rewritten and improved the discussion section, with more information been added in the revised manuscript. Furthermore, we established the close associated relationship among morphogenic potential, corresponding histochemical and cytological characteristics. And the main differences between the analyzed calli have been proposed and efficiently identified in our previous research publication as mentioned above (Guo et al. 2019). The representative staged samples which are feasibly identified by the morphological characteristics, and would be highly enriched and collected in our study. Finally, as the reviewer suggested, the conclusion section has been rewritten and added in the revised manuscript.
Reviewer 2 Report
The MS appears too long and how to say it presents a weak logic that fails to highlight the importance of the many results. In fact, Authors fail to present which comparison could be more important, e.g. PEC vs. NEC and/or GE vs. PEC. Personally, I believe that the comparison GE vs. NEC is the less appropriate of the three.
Moreover:
Line 59: “today” for a paper of 2005 is excessive;
Lines 107-117: Please add references;
Figure 5: the format of Figure 4 must be maintained because a Venn diagram between three comparisons is very difficult to understand;
Table 2: it is not clear to what comparison it refers;
Table 3: it is not clear to what comparison it refers;
Figure 7: Authors present only two comparison, why? (metabolites and not metabolin
Table 4: please check the legend;
Table 5: it is not clear to what comparison it refers;
Abbreviations must be listed in alphabetic order
Conclusions are necessary to summarize the genes /metabolites candidate for a relevant role in SE trans-differentiation.
Author Response
Dear Editor and Reviewer,We are returning our manuscript (ijms-473380) entitled “Metabolome and transcriptome association analysis reveals dynamic regulation of purine metabolism and flavonoid synthesis in transdifferentiation during somatic embryogenesis in cotton” that has been revised based on the reviewer's suggestions. We appreciate the time and expertise of the reviewer and the suggestions helped us to improve our manuscript.
Here are our point by point response to the reviewer's comments:
Reviewer 2
Comments and Suggestions for Authors:
1. The MS appears too long and how to say it presents a weak logic that fails to highlight the importance of the many results. In fact, Authors fail to present which comparison could be more important, e.g. PEC vs. NEC and/or GE vs. PEC. Personally, I believe that the comparison GE vs. NEC is the less appropriate of the three.
RESPONSES: We thank and agree with the reviewer’s valuable comments and suggestions. As the reviewer recommend, the manuscript has been revised accordingly to make the result highlighted and more concise. We have focused on comparing the changes of me tabolites/genes abundance between the neighbor stages, PEC vs. NEC and GE vs. PEC (like in the association analysis of transcriptome and metabolome), to identify the significant differential accumulation of metabolites/genes between the two SE transdifferentiation processes, embryogenic differentiation (PEC vs. NEC) and somatic embryo developmental initiation (GE vs. PEC).
2. Line 59: “today” for a paper of 2005 is excessive.
RESPONSES: We thank and agree with the reviewer for the comments and suggestions. The term “today” has been deleted in the revised manuscript.
3. Lines 107-117: Please add references.
RESPONSES: We thank the reviewer’s comments and suggestions. As the reviewer recommend, the relevant references have been cited in the revised manuscript
4. Figure 5: the format of Figure 4 must be maintained because a Venn diagram between three comparisons is very difficult to understand.
RESPONSES: We thank and agree with the reviewer’s comments and suggestions. In Figure 5, the Venn diagrams between two comparisons have been fixed, maintaining the format of Figure 4.
5. Table 2: it is not clear to what comparison it refers.
RESPONSES: We thank the reviewer for pointing out the confused description here. The comparison “PEC vs. NEC” has been added in Table 2.
6. Table 3: it is not clear to what comparison it refers.
RESPONSES: We thank the reviewer for pointing out the confused description here. The comparison “GE vs. PEC” has been added in Table 3.
7. Figure 7: Authors present only two comparison, why? (metabolites and not metabolin).
RESPONSES: We thank the reviewer for the valuable comments and suggestions. We presented only two comparisons in Figure 7, in order to focus on comparing the significant differential accumulation of metabolites/genes between the two critical SE transdifferentiation processes, embryogenic differentiation (PEC vs. NEC) and somatic embryo developmental initiation (GE vs. PEC). Besides, we agree with and implement the reviewer’s comments that “metabolites” instead of “metabolin” should be used in Figure 7.
8. Table 4: please check the legend.
RESPONSES: We thank the reviewer’s comments and suggestions. The legend of Table 4 has been rewritten to make it clear as recommend.
9. Table 5: it is not clear to what comparison it refers.
RESPONSES: We thank the reviewer for pointing out the confused description here. The comparison “PEC vs. NEC” has been added in Table 5.
10. Abbreviations must be listed in alphabetic order.
RESPONSES: We thank and agree with the reviewer’s comments. As suggested, the abbreviations have been listed in alphabetic order in the revised manuscript.
11. Conclusions are necessary to summarize the genes/metabolites candidate for a relevant role in SE trans-differentiation.
RESPONSES: We acknowledge and agree with the reviewer for the comments and suggestions. The conclusion section summarizing the genes/metabolites candidate for a relevant role in SE trans-differentiation has been rewritten and added in the revised manuscript as the reviewer recommend.
We hope you find the revised manuscript satisfactory and thank you for considering our manuscript.
Sincerely,
Fanchang Zeng
Professor
State Key Laboratory of Crop Biology
College of Agronomy,
Shandong Agricultural University
Tai’an 271018
Phone: +86-538-8241828
E-mail: fczeng@sdau.edu.cn
Round 2
Reviewer 1 Report
Authors improved the whole manuscript.
However, before it will be published:
Line 346 ‘during SE in cotton’ is in italics. Correct it.
Table 4 You have the names of the genes so it have to be in italics!
Line 478-481 something wrong with the text formatting, everything is in italics.
Authors contributions parts have to be changed from the whole names to only the first letter of the names and surnames. I think this is the journal requirements.
Author Response
We thank the reviewer for pointing out the formal mistakes there. All the sentences have been revised to be correct accordingly. Simultaneously, we also checked and corrected other formal mistakes in the whole manuscript.
Reviewer 2 Report
The only suggestion is an English revision to facilitate the reading